# The Toxic Mechanism of Gliotoxins and Biosynthetic Strategies for Toxicity Prevention

**DOI:** 10.3390/ijms222413510

**Published:** 2021-12-16

**Authors:** Wei Ye, Taomei Liu, Weiyang Zhang, Weimin Zhang

**Affiliations:** 1Guangdong Provincial Key Laboratory of Microbial Culture Collection and Application, State Key Laboratory of Applied Microbiology Southern China, Institute of Microbiology, Guangdong Academy of Sciences, Guangzhou 510070, China; yewei@gdim.cn (W.Y.); liutm@gddcm.com (T.L.); zhangwy@gdim.cn (W.Z.); 2Department of Ocean Science and Hong Kong Branch of Southern Marine Science and Engineering Laboratory (Guangzhou), Hong Kong University of Science and Technology, Hong Kong, China

**Keywords:** gliotoxin, toxic mechanism, NF-κB, detoxification, biosynthetic strategies

## Abstract

Gliotoxin is a kind of epipolythiodioxopiperazine derived from different fungi that is characterized by a disulfide bridge. Gliotoxins can be biosynthesized by a *gli* gene cluster and regulated by a positive GliZ regulator. Gliotoxins show cytotoxic effects via the suppression the function of macrophage immune function, inflammation, antiangiogenesis, DNA damage by ROS production, peroxide damage by the inhibition of various enzymes, and apoptosis through different signal pathways. In the other hand, gliotoxins can also be beneficial with different doses. Low doses of gliotoxin can be used as an antioxidant, in the diagnosis and treatment of HIV, and as an anti-tumor agent in the future. Gliotoxins have also been used in the control of plant pathogens, including *Pythium ultimum* and *Sclerotinia sclerotiorum.* Thus, it is important to elucidate the toxic mechanism of gliotoxins. The toxic mechanism of gliotoxins and biosynthetic strategies to reduce the toxicity of gliotoxins and their producing strains are summarized in this review.

## 1. Introduction

Gliotoxin is a kind of epipolythiodioxopiperazine (EPT) characterized by a disulfide bond, which is produced by various kinds of fungi, including *Aspergilluss fumigatus* [1,2], *Trichoderma virens* [3], and *Dichotomyces cejpiii* [4,5]. Gliotoxin was firstly isolated from *A. fumigatus*, which exhibited cytotoxic effects via antiangiogenesis, anti-inflammation, immunosuppression, apoptosis, production of ROS, and genotoxicity [6]. Gliotoxin isolated from *Trichoderma virens* was also reported to inhibit the growth of plant pathogens via the suppression of phagocytosis and the killing of conidia of versatile pathogenic fungi [7,8]. Gliotoxin can be prepared as a fungicide and insecticide to control plant diseases, especially for rice blight, rice blast, and rice stunt. Gliotoxin-producing strain *Trichoderma virens* has been developed into other preparations, including GL-21, Glio Gard^TM^ and Soil Gard^TM^, to control seedling diseases and root diseases of greenhouse and field crops with remarkable effects [9,10]. Gliotoxin can also inhibit tumor cell proliferation and viral RNA replication and induce apoptosis of hepatic stellate cells as well as hinder the expression of nuclear factor-kappa B (NF-κB), exhibiting promising prospects in the fields of antitumor, antivirus, and anti-liver fibrosis and immunosuppression during organ transplantation [6]. Gliotoxin has been demonstrated to repress antigen-presenting cell function and induce the apoptosis of monocytes, thus inducing immuno-evasive effects on T-cells and antigen-presenting cells [11]. Gliotoxin could also alleviate the development of hepatic fibrosis and cirrhosis in a thioacetamide rat model via the induction of apoptosis of activated hepatic stellate cells in the liver [12]. In addition, gliotoxin showed advantages in its ability to easily penetrate and impair the blood–brain barrier because of the interaction of gliotoxin with membrane proteins [13], suggesting the therapeutical potential of gliotoxin in different doses.

At present, there is a large gap in the market for gliotoxins due to their complex chemical structure and insufficient yield. Moreover, the toxic mechanism of gliotoxin with different doses and their biosynthetic prevention strategies remain obscure, thus limiting the wide application of gliotoxins and their derivatives in the biomedical and agriculture industries. Therefore, a review of the toxic and detoxification mechanisms of gliotoxins would provide clues for the future broad application of gliotoxins and their derivatives as well as reduce the hazard of the gliotoxin-producing strain *Aspergillus fumigatus.*

## 2. The Underlying Toxic Mechanism of Gliotoxins

### 2.1. The Biosynthesis of Gliotoxin

Gliotoxin is the first isolated ETP with the most simple structure; more than 100 kinds of gliotoxin-like compounds have been excavated, such as chaetocin, chetomin, and sporidesmin. Gliotoxin was firstly isolated in *Trichoderma virens* [7] and then was found in other fungi, including *A. fumigatus* [2], *Aspergillus niger*, *Aspergillus terreus*, *Eurotium chevalieri,* and other *Penicillium* and *Acremonium* species [14]. The biosynthetic mechanism of gliotoxin mediated by the *gli* gene cluster in *A. fumigatus* has been elucidated [15,16]. The non-ribosomal peptide synthetase encoded by the *gliP* gene catalyzes the diketopiperazine scaffold formed by the non-ribosomal peptide synthetase (GliP); then, the hydroxyl group is added to the amino acid by cytochrome P450 monooxygenase (GliC), followed by the sulfurization catalyzed by glutathione S-transferase (GliG). The sulfur bond is important for the toxicity of gliotoxins. Next, the -glutamyl moieties are cut off by the catalyzation of GliK and GliJ, and a notorious epidithiol moiety is then formed, intermediated by *gliI-*encoding amino transferase, the N-methyltransferase (GliN) encoded by *gliN* and/or O-methyltransferase GliM encoded by *gliM,* which play the role of freestanding amide to accelerate amide methylation and confer stability on the ETP. Additionally, gliotoxin is generated by the formation of a disulfide bond catalyzed by oxidoreductase (GliT). GtmA, which is not in the gliotoxin biosynthesis gene cluster, can intermediate the cleavage of the disulfide bond in the gliotoxin to form bisdethiobis(methylthio)-gliotoxin; moreover, GliA is responsible for pumping out the gliotoxin to the outer membrane; therefore, both GtmA and GliA can attenuate the cytotoxicity of gliotoxin to the host, thus proving an effective strategy to alleviate the hazard and cytotoxicity of gliotoxin to different hosts [16]. The biosynthetic pathway for the biosynthesis of gliotoxin and its derivatives is proposed in Figure 1. Moreover, GliZ has been demonstrated as a positive regulator for the biosynthesis of gliotoxins; the disruption of *gliZ* would abolish the production and gliotoxin [17]. Moreover, LaeA [18] and VeA [19] global regulators have also been demonstrated to play an important role in the generation of gliotoxins in *A. fumigutas*. In addition, the in vitro function of genes related to gliotoxin biosynthesis, including *gliZ*, *gliG*, *gliI*, and *gliO* in deep-sea fungus *Dichotomyces cejpii* FS110 [20] and gliotoxin biosynthesis-related genes *gliK*, *gliM,* and *gliT* in deep-sea fungus *Geosmithia pallida* FS140 [21], have been characterized by their biochemical reaction. Interestingly, GliK was reported to exert the function of gliotoxin acetylation, thus facilitating the generation of gliotoxin derivates in *G. pallida* FS140. The function of GliF (monooxygenase) and GliH in the *gli* gene cluster has remained obscure until now, which needs further investigation.

Gliotoxin and its derivatives have been excavated from deep-sea fungus *D. cejpii* FS110 [20] and *G. pallida* FS140 [21] in our group. Additionally, epipolythiodioxopiperazine derivatives of gliotoxin with the dithiol group showed weaker cytotoxicity compared with gliotoxin, and gliotoxin derivates of the methylthio group showed much weaker cytotoxicity than that of dithiol gliotoxin. Furthermore, the depletion of the methythio group in gliotoxin derivative leads to the loss of cytotoxicity, suggesting the important role of the disulfide bridge and thiol group in the biological activity of gliotoxin and its derivatives.

### 2.2. The Underlying Toxic Mechanism of Gliotoxin

Gliotoxin can induce cytotoxicities via different mechanisms, including the inhibition of angiogenesis, immunosuppressive activity, the inducement of cell apoptosis, the suppression of the activity of protease, the production of ROS and the inhibition of the enzymatic activity of peroxidases, the imbalance of oxireductase reaction, the induction of DNA damage, and the repression of specific genes’ expression, including NF-κB factor [6].

#### 2.2.1. The Inhibition of Angiogenesis

Previous studies have demonstrated that gliotoxin can inhibit angiogenesis in humans with the symptom of invasive aspergillosis, which was intermediated by the gliotoxin produced by *A.*
*fumigatus* [22]. Meanwhile, invasive aspergillosis is the most common infected fungal disease that causes high mortality, especially for immune deficiency patients, because of the deficiency in the approaches for diagnosis and treatment [23]. Additionally, the incidence rate of invasive aspergillus is increasing year by year due to the wider application of organ transplants and antibiotics [23]. As a consequence, it is important to elucidate the mechanism of aspergillosis mediated by gliotoxin. Gliotoxin can repress angiogenesis in the lung of mice and patients suffering from aspergillosis; furthermore, *A. fumigatus* could impede capillary tube formation in a murine model of cutaneous infection via the generation of secondary metabolites, especially gliotoxins [24]. Additionally, pure gliotoxin can also suppress angiogenesis and endothelial cell migration in vitro [22,25]. The deletion of *gliP* in the biosynthetic gene cluster and LaeA global transcriptional regulator can significantly attenuate the angiogenic effect caused by *A. fumigatus* [22,26], thus further supporting the function of angiogenesis caused by gliotoxin [27]. In addition, the immune response can also be inhibited via the antiangiogenic effect mediated by gliotoxin, thus further deteriorating the symptom of invasive aspergillus.

The proangiogenic and antianogentic effects caused by *A. fumigatus* has been investigated. The antiangiogenesis effect of gliotoxin produced by *A. fumigatus* was exerted via the reduction of the generation of ROS and, thereby, the suppression of NF-κB [28]. Furthermore, gliotoxin can also repress the proliferation of polymorphonuclear leucocytes [29]. Moreover, gliotoxin can suppress the generation of NF-κB, thus blocking the production of cytokines and inflammatory factors to reduce the angiogenic effect [30]. To compensate the antigiogenic effect, thus protecting the host, the infection with *A. fumigatus* triggers proangiogenic effects, including the production of proinflammatory factors such as tumor necrosis factor alpha (TNFα), hypoxia-inducible factor 1-alpha (HIF1α), and interleukin (IL-8), thus recruiting the polymorphonuclear leucocytes (PMNLs) and leading to the burst of ROS, including H_2_O_2_. The gene ratio of ROS can induce the upregulation of NF-κB, thus elevating the expression of vascular endothelial growth factor and other proangiogenic factors to compensate the antanginogenic effect caused by *A. fumigatus* [22,31]. Moreover, gliotoxin can abrogate the angiogensis induced by H_2_O_2_ in low concentrations, and gliotoxin can completely inhibit the angiogenic effect in immunosuppressive mice treated with cyclophosphamide via the abolishment of the production of ROS, which is mediated by gliotoxin-catalyzed reduction by the assistance of NAPDH [32]. Therefore, these studies also provide strategies for the prevention of angiogenic effects of invasive aspergillus via the upregulation of proangiogenic mediators, including NF-κB, cytokines, and vascular endothelial growth factor (VGEFs). Different concentrations of gliotoxins can induce different proangiogenic and antiangiogenic effects; thus, the angiogenic effect can be balanced via the control of gliotoxin concentrations.

#### 2.2.2. Immunosuppressive Activity

Gliotoxin has been reported to induce immunosuppressive activity via cell apoptosis mediated by the repression of phagocytosis and macrophages [33] as well as the suppression of the proliferation of giant cells and T-cells [33]. Moreover, gliotoxin can further deteriorate the diseases caused by fungi infection via the inhibition of the activities of immune-related cells [33,34]. The amounts of immune cells, including leucocyte, lymphocyte, and Langerhans cells, significantly decreased after the treatment with gliotoxins in adult camels for 7 days. Furthermore, gliotoxin from *A. fumigatus* abolish the generation of oleukotriene B4 through the suppression of leukotriene A4 hydrolase, thus exerting the function of immunosuppression [35]. In addition, the knockout of *gliP* and *gliZ* genes attenuated the toxicity of gliotoxin towards nonneutrophil cells in mice [36]; 50 nM of gliotoxin can exert immunosuppressive activity via the suppression of the phagocytosis of astrocytes [24,37], which led us to exploit low-dose gliotoxins as immunosuppressive agents during an organ-transplant operation, thus reducing immunological rejection in organ transplant patients.

The immunosuppressive mechanism of gliotoxin has been widely investigated. It is well known that gliotoxin can inhibit the activation of NF-κB mediated by the repression of IκBα degradation caused by the chymotrypsin-like activity of the 20S proteasome [38]. The disulfide of glitoxin can target the protease, which can be reversed by reducing agent dithiothreitol, which can be reduced from gliotoxin to dithiol gliotoxin. Low doses of gliotoxin, with a concentration of 1 μM, exhibited 63% inhibition of the LLVY-amc peptide-hydrolyzing activity mediated by the 20S proteasome, suggesting the high immunosuppressive efficiency of gliotoxin [38].

NF-κB is a kind of inducible transcriptional factor; some examples are NF-κB1 (p50), NF-κB2 (p52), RelA, RelB, and c-Rel. A typical active NF-κB is the P50-P65 dimer that can bind DNA; thus, activating NF-κB will modulate inflammation and innate and adaptive immune system reactions. The IκBα protein can inhibit the activity of NF-κB in the cytoplasm via binding with NF-κB, which can be activated by the degradation of the IκBα protein, mediated by the 20S protease, and by the binding of the DNA in the nucleus [39,40,41]. Glitoxin can stabilize the expression level of IκBα via the inhibition of the 20S protease, blocking the activation of NF-κB and thus exert the function of immunosuppression; the disulfide bond in gliotoxin is essential for its immunorepression activity [38]. In consequence, the disruption of *gliP* can abolish the immunosuppression function of *A. fumigatus* via the suppression of gliotoxin production [42]. Furthermore, gliotoxin can assist *A. fumigatus* in achieving immune escape from the phagocytosis of macrophages by targeting the metabolism of phosphatidylinositol 3,4,5-trisphosphate [PtdIns(3,4,5)P3] [43]; the gliotoxin can promote the immune escape of *A. fumigatus* by decreasing the content of intracellular lipid acyl inositol 3,4,5-trisphosphate [43], thus leading to the function deficiency of the integrin and actin backbone and preventing cell membrane protrusion extension [44], further impairing the function of macrophages. Furthermore, it was reported that gliotoxin contributed to the invasion of *A. fumigatus* spores into cells via the activation of phospholipase D in lung epithelial cells and by inducing actin backbone rearrangement [45,46], but the specific regulatory mechanism remains obscure.

#### 2.2.3. Inflammatory and Anti-Inflammatory Effects

Gliotoxin can also initiate the production of inflammatory factors, thus eliciting further cytotoxicity. Gliotoxin incubation with A459 cells can induce the secretion of proinflammatory cytokines, including IL-4, IL-8, and IL-10, whereas the upregulation of IL-6 and IL-8 cytokines was observed in gliotoxin-treated L132 cells, with only IL-8 upregulation observed in A459 and L132 cells treated with fumagillin [34,47]. In addition, the expression of a few cytokines, such as IL-1β, IL-17, interferon-gamma (IFN-γ), TNF-α, and granulocyte macrophage-colony stimulating factor (GM-CSF), was downregulated in the two cells treated with gliotoxin and fumagillin, suggesting neutrophil-mediated tissue damage after *A. fumigutas* infection [47]. The results also suggest a stronger inflammation-eliciting effect by gliotoxin compared with fumagillin. Gliotoxin was administrated to female C57BL/6 mice immunized with myelin oligodendrocyte glycoprotein with a dose of 1 mg/kg, which can aggravate the symptom of neuroinflammation via the upregulation of inflammatory genes, including T-box transcription factor (TBX21), inducible nitric oxide synthase (iNOS), arginase 1 (ARG1), and cytokines such as IFN-γ, IL-17, and IL-2 [48].

In contrast, in vivo anti-inflammatory activity was also investigated. Herfarth et al. reported that gliotoxin can suppress intestinal inflammation in dextran sulfate sodium (DSS)-induced colitis mediated by NF-κB activation [49]. The expression of cytokines, including TNF-α, was downregulated in RAW-264.7 mouse macrophage-like cells; meanwhile, TNF-α and IL-1α mRNA expressions were also significantly suppressed in DSS-induced colitis mice by gliotoxin treatment for 8 days, thus leading to the alleviation of the symptom of colonic inflammation in mice [49]. Furthermore, the gliotoxin treatment also reduced the DNA-binding activity of NF-κB, thus impeding the inflammation mediated by NF-κB activation in the colon [30]. In addition, gliotoxin can ameliorate trinitrobenzene sulfonic acid (TNBS)-induced mouse colitis via the suppression of the expression levels of TNF-, IL-1, and intercellular adhesion molecule-1 (ICAM-1) proteins as well as the upregulation of heme oxygenase-1, thus indicating the potential application of gliotoxins in the clinical treatment of Crohn’s disease [50] via the induction of hemooxygenase-1, which can oxidize low-density lipoproteins with the potential to form chemo-attractants, thus alleviating the inflammation [51].

In summary, gliotoxin’s toxic effects, including the antiangiogenesis and inflammatory-related effects mediated by NF-κB activity and redox cycling, are summarized in Figure 1. Gliotoxin can target the 20S proteasome, which is responsible for the degradation of the IK-Bα protein that specifically binds to NF-κB; thus, gliotoxin can suppress NF-κB activity. NF-κB is closely associated with the function of angiogenesis, immune response, and cell proliferation; thus, gliotoxin treatment can induce the effects of immunosuppression, antiangiogenesis, and anti-inflammation (Figure 2).

#### 2.2.4. Inducing the Production of ROS and the Inhibition of Peroxidase

Gliotoxin can induce the production of reactive oxygen species (ROS) via intracellular redox cycling [52], thus resulting in damage from the peroxidation of macromolecules and biofilms, mediated by oxidative stress and the abrogation of antioxidation. In human neutrophils, reduced gliotoxin acts as a generator of superoxide by inhibiting NADPH oxidase [53]. Alternatively, gliotoxin can also served as an antioxidant. Reduced nicotinamide adenine dinucleotid phosphate (NADPH) peroxidase can only produce low contents of ROS to scavenge pathogenic bacteria and fungi, thus protecting the hosts. Meanwhile, Tsunawaki et al. reported that gliotoxin can inhibit the enzymatic activity of NADPH peroxidase with a concentration of higher than 1 μg/mL, thus causing damage to the host, whereas the NADPH peroxidase in neutrophils was not inhibited by the treatment with this dose of gliotoxin. Thus, different doses of gliotoxin induce different oxidative damage and different physiological effects; 30–100 ng/mL gliotoxin can inhibit the phagocytosis of zymosan without affecting ROS production. Gliotoxin can also lead to the reshaping of the actin cytoskeleton [54], thus contributing to cell shrinkage and filopodia disappearance, which can be reversed by Cyclic Adenosine monophosphate (cAMP) antagonist Rp-cAMP; however, gliotoxin-induced phagocytosis can not be inversed by cAMP [54].

The detailed mechanism of the gliotoxin-induced mechanism mediated by ROS has been investigated. Bernardo et al. described a redox-uptake system mediated by gliotoxin [55]: intracellular gliotoxin can be reduced to dithiol gliotoxin by a cellular reducing agent, thus releasing electrons to cellular O_2_ to form O_2_-mediated gliotoxin, which can yield ROS. Furthermore, dithiol gliotoxin can also be oxidized into gliotoxin by cellular O_2_ in turn, thereby resulting in the accumulation of ROS via the redox-uptake system mediated by gliotoxin. Thus, gliotoxin can cause cell peroxide damage mediated by the redox-uptake system in a dose-dependent manner.

On the other hand, gliotoxin can also be explored as an antioxidant. Gliotoxins exhibited the ability to reduce H_2_O_2_ to H_2_O in a thioredoxin redox system composed of thioredoxin, thioredoxin reductase, and NADPH [56,57], whereas gliotoxin did not show H_2_O_2_-reducing activity in a glutathione (GSH) / glutathione disulfide (GSSG) reductase system. The H_2_O_2_-reducing activity in the yeast Trx system depends on the concentrations of gliotoxin and H_2_O_2_, which was applicable to the rat Trx system [57], suggesting gliotoxin as a potent antioxidant for mammalian cells. The in vitro H_2_O_2_-reducing activity of gliotoxin in Hela cells was also investigated; the results indicated that the H_2_O_2_ level was significantly decreased by the addition of gliotoxin, with concentrations ranging from 40–160 nmol, in a dose-dependent manner in Hela cells, which was demonstrated by 2′,7′-dichlorofluorescein diacetate (DCFH-DA) probes [57]. In addition, gliotoxin can target NADPH oxireductase to prevent the onset of O_2_^-^ generation in human neutrophils [58] and lead to the release of cytochrome c through a reaction with intracellular reductants in an oxygen-dependent manner; the prevention of redox cycling facilitated the damage of gliotoxin to NADPH oxireductase with IC_50_ of 9 nM [59]. Furthermore, the detailed targeting of gliotoxin to the NADPH oxidase system was investigated by a cell-free activation assay with NADPH oxidase components. The ROS generation ability of the membrane was reduced by 40% after gliotoxin treatment [29,59,60], and the in vitro addition of gliotoxin to flavocytochrome b558 and cytosolic components also exhibited an inhibition rate of 50% with the concentration of 3.3 μM [60], suggesting that flavocytochrome b558 is the target of gliotoxin. Moreover, the cell surface of alkaline phosphatase activity, which is a marker enzyme of oxidant-producing intracellular compartments (OPICs), is also suppressed by gliotoxin in human neutrophils stimulated with phorbol myristate acetate (PMA) as well as the repression of NADPH oxidase [61], thus contributing to the inhibition of superoxide production in PMA-treated human neutrophils.

Moreover, the block of oxidative stress by gliotoxin would impede the proliferation of human umbilical vein endothelial cells in a dose-dependent manner with the IC_50_ of gliotoxin of 250 nM [56,62], which was attributed to the fact that oxidative stress can trigger angiogenesis. Nevertheless, gliotoxin exhibited cytotoxicity to human umbilical vein endothelial cells (HUVECs) with concentrations higher than 500 nM [56,62], whereas there was no obvious cytotoxicity with concentrations lower than 300 nM, which is consistent with a previous study that found that gliotoxin induces apoptosis in activated human hepatic stellate cells with concentrations ranged from 0.3 to 7.5 μM [63]. To further elucidate the role of H_2_O_2_ and gliotoxin in angiogenesis, 100 nM H_2_O_2_ was added to induce the sprout of endothelial cells and the formation of thick tubes, which were inhibited by the addition of gliotoxin with concentrations of 125 to 250 nM [62]. In addition, gliotoxin potently inhibited the H_2_O_2_-stimulated invasion of HUVECs to the control level at 125 nM [62]. Thus, gliotoxin can exert the function of antiangiogenesis via the regulation of the intracellular redox state. The thioredoxin-SH group can reduce the gliotoxin to dithiol gliotoxin while leading to the reduction of H_2_O_2_ to H_2_O [56,64,65], thereby alleviating the angiogenic and invasive effects mediated by H_2_O_2_. Therefore, low doses of gliotoxin show great potential to be developed as an antioxidant to encounter the peroxide damage caused by H_2_O_2_.

Taken together, the ROS-inducing effects mediated by the redox cycling system caused by gliotoxin are summarized in Figure 3. Gliotoxin can be reduced to dithiol gliotoxin by in vivo reducing agents, including dithiothreitol (DTT), glutathione, and reducing enzymes, thus providing electrons to oxygen atoms and, thereby, producing intracellular ROS. Excessive ROS can cause peroxide damage and single- and double-strand breaks, causing great damage to the human body. In contrast, gliotoxin can be served as an antioxidant in the presence of the thioredoxin redox system through the inhibition of NADPH oxidase. Moreover, both gliotoxin and ditholgliotoxin are cytotoxic; however, gliotoxin is more toxic, probably due to the existence of the disulfide bridge, which can bind to many physiological-related proteins.

#### 2.2.5. Genotoxicity of Gliotoxin

Gliotoxin can also cause genotoxicity mediated by DNA damage in vitro and in vivo. Ditholgliotoxin can cause single- and double-stranded breaks in the presence of DNA and Fe^3+^, which was determined by neutral agarose gel electrophoresis [66,67]. Gliotoxin did not induce single- and double-stranded breaks except by the addition of reducing agents, including glutathione, DTT, and pyridine reducing enzymes [67], suggesting the genotoxicity was probably mediated by ditholgliotoxin and this DNA damage effect can be abrogated by metal chelator and catalase. This supports that the DNA damage effect was mediated by the ROS produced during the redox cycling process to form hydroxylated and other altered DNA products, which was determined by 32P DNA radiolabelling and two-dimensional thin-layer chromatography [68].

DNA damage was observed in mouse RAW264.7 macrophages after gliotoxin treatment for 2 h in plain medium in a single cell electrophoresis assay [67,69,70,71]; however, a clear, dose-related increase in sister-chromatid exchange was not observed in Chinese hamster ovary (CHO) cells [72]. Gliotoxin can also cause mitochondrial damage, thus producing ROS to induce DNA damage and causing toxicity to lung epithelial cells A459 and L132; this is stronger than another toxin fumagillin produced by *A. fumigatus* [47]. It is also speculated that gliotoxin can induce lung epithelial cells damage mediated by endoplasmic reticulum (ER) stress due to the observation of the accumulation of unfolded proteins in the lumen of the ER. Additionally, the cytotoxicity of gliotoxin towards different organ-derived cells is as follows: renal epithelial cells > type II epithelial cells > hepatocytes > normal lung epithelial cells [47], indicating that gliotoxin has selective cytotoxicity towards organ-derived cells to some extent.

#### 2.2.6. The Induction of Cell Apoptosis

Gliotoxin can cause cell damage via the induction of apoptosis. It was confirmed that gliotoxin has strong cytotoxicity on thermosensitive mouse breast cancer cells [73,74]. To investigate the role of gliotoxin in brain injury induced by *Aspergillus*, astrocyte and nerve cells were cultured with gradually increasing gliotoxin concentration; the results showed that gliotoxin at 300 and 1000 nM reduced the mitochondrial activity of astrocyte cells and neurons at 18 and 5 h, respectively [75]. It was found that gliotoxin could also induce the cytotoxicity of small nerve cells [75]. Studies have also shown that the toxicity of gliotoxin to neutrophils occurs by damaging the flavocytochrome b558 ion conversion before inhibiting the activity of the oxidase [60]. The IC_50_ of gliotoxin towards HT-29 colorectal cancer cells is 0.6 μg/mL [76]. Apoptosis was observed in nerve cells after the treatment with 300 nM gliotoxin for 12 h [75]. It was also demonstrated that gliotoxin-producing strain *A. fumigutas* can induce apoptosis in human bronchial epithelial cells via the upregulation of Bak and the downregulation of Bcl-2 [77,78], while the deletion of the *gliP* gene can abrogate this apoptotic effect [79].

Gliotoxin can induce apoptosis via different mechanisms. The phosphorylation of histone H3 on Ser-10 was observed in thymocytes after treatment with gliotoxin for 10 min [80], resulting in apoptosis mediated by the upregulation of cyclic AMP levels and protein kinase A (PKA) activity, which can be repressed by PKA inhibitors, including genestein [80]. Additionally, the sensitivity of chromatin to nuclease digestion was also enhanced by the phosphorylation of histone H3, leading to further apoptosis [80]. In human cervical cancer (Hela) and human chondrosarcoma (SW1353) cells, gliotoxin induces apoptosis mainly through the mitochondrial pathway, which induces the activation of caspase-3, caspase-8, and caspase-9 and downregulates the expression of Bcl-2 and upregulates Bax expression and cytochrome c (cyt c) release [81]. In mice fibroblasts, human bronchial epithelial cells, and mouse alveolar epithelial cells, gliotoxin and the supernatant of *A. fumigatus* activated the JNK pathway, and JNK mediated the triple phosphorylation (S100, T112 and S114) of Bimel-, bak-, and caspase-dependent apoptosis [82]. This signaling pathway appears to be a powerful tool for *A. fumigatus* to kill lung epithelial cells in a gliotoxin-dependent manner, and this mechanism may also explain why gliotoxin has a role in the development of invasive aspergillosis. Gliotoxin induces apoptosis in many mammalian cells via the JNK-mediated BIM phosphorylation signaling pathway, which provides us with a new understanding of BIM activation [82]. According to the phenotypes of various gene-deficient mouse strains, BIM is the most important BH3-only protein in the immune system and participates in the apoptosis of many different cell types. The BH3-only protein is considered a sensor of the mitochondrial apoptosis pathway, and it is the upstream component of the Bcl-2 family to regulate mitochondrial apoptosis [83]. Gliotoxin induces apoptosis in hematopoietic stem cells and human hepatic stellate cells through oxidative stress mediated by ROS, and ROS also accelerates the release of mitochondrial cytochrome c and apoptosis-inducing factors, thereby promoting apoptosis. In addition, other studies have shown that giotoxin was released in an oxidative form after inducing intracellular apoptosis and entered a neighboring cell to induce apoptosis, which suggests that gliotoxin can induce apoptosis with just a tiny amount [84]. Moreover, glitoxin with a concentration of 1.5 µM could trigger the apoptosis of rat activated hepatic stellate cells via interaction with adenine nucleotide transporter (ANT), which is a protein located on the inner mitochondrial membrane that carries adenosine diphosphate and adenosine triphosphate across the membrane. The disruption of ANT by gliotoxin can alter the mitochondrial membrane potential [85].

Moreover, the novel apoptosis mechanism mediated by gliotoxin has also been explored recently. Gliotoxin can target integrins to modify their cysteines by a disulfide bond, thus blocking the binding of arginine-glysine-aspartic acid (RGD)-containing extracellular matrix, leading to the inhibition of focal adhesion kinase and the dephosphorylation of p190RhoGAP, which allows for the activation of RhoA and, subsequently, the activation of ROCK, MKK47/MKK7, and JNK and the induction of Bim phosphorylation, which is a proapoptotic factor. Finally, the treatment of gliotoxin in BEAS-2B adherent cells leads to anoikis [44,86], which is a form of apoptosis mediated by cell attachment. In addition, reduced gliotoxin (dithiol gliotoxin) can also trigger the rapid cell detachment of colorectal cancer cells (CRC) via the disruption of integrin binding to RGD, thus inducing anoikis in CRCs. Moreover, reduced gliotoxin can also initiate the generation of excessive ROS and the disruption of membrane potential, thus further promoting apoptosis in CRCs [86]. It was also demonstrated that the exposure of lung epithelial cells A549 and L132 to gliotoxin for 24 h resulted in a significant increase in the proportion of cells in S-phase (30–39%), with a concomitant decrease in G2/M-phase; the early and late apoptotic cells were observed using Annex, PI stains, suggesting apoptosis in the two cells after gliotoxin treatment [47]. In addition, the expression levels of antiapoptotic Bcl-2 and caspase-8 decreased, whereas expression levels of BAK, BAX, BID, and caspase-3 were significantly increased in both A549 and L132 cells, suggesting gliotxin induce the apoptosis of A459 and L132 cells via the activation of caspases and the Bax pathway, also implicating mitochondrial damage [47]. Gliotoxin can also induce the apoptosis of chronic lymphocytic leukemia cells (CLLs) (IC_50_ between 0.1 and 1 μM), associated with the inhibition of the NOTCH2/FCER2 (CD23) axis, together with the upregulation of the NOTCH3/NR4A1 axis and NF-κB factor [87,88], thus providing a novel strategy for the treatment of CLLs. In addition, gliotoxin can also induce apoptosis via the deregulation of NOTCH2 mRNA expression towards cell lines derived from melanoma (518A2), hepatocellular carcinoma (SNU398, HCC-3, Hep3B), and pancreas carcinoma (PANC1) with the high expression level of NOTCH_2_; the apoptotic effect was not observed in the NOTCH-negative cell line Huh7 [87], suggesting the vital role of NOTCH in gliotoxin-inducing apoptosis. Meanwhile, gliotoxin can significantly reduce the tumor volume of melanoma xenograft mice [87].

Surprisingly, gliotoxin can also induce apoptosis in paclitaxel-resistant ovarian cancer cells via the upregulation of active p63 (TAp63), and it was demonstrated that the pretreatment of ovarian cancer cells with gliotoxin can inhibit the expression of drug-resistant proteins MDR1 and MRP1-3, thus inducing caspase-dependent apoptosis through autophagy after subsequent treatment with paclitaxel [89]. These results suggest that gliotoxin has the potential application of killing chemoresistant cells. Moreover, gliotoxin also triggered the apoptosis of adriamycin-resistant (ADR) non-small-cell lung cancer cell (NSCLC) lines A459/ADR (IC_50_ values of 0.40 and 0.24 μM) via the disruption of mitochondrial membrane potential and the upregulation of p53, p21, Bax, cleaved poly (ADP-ribose) polymerase (PARP) and caspase-9 [90], suggesting the promising prospect of gliotoxin in ADR NSCLC. Additionally, gliotoxin induced apoptosis via different signal pathways; this is summarized in Figure 4.

#### 2.2.7. Other Effects Induced by Gliotoxin

Gliotoxin can also target other proteins rich in cysteine and induce other physiological effects due to the presence of the disulfide bond in gliotoxin. The inhibitory effect of gliotoxin on heat shock protein activity was observed by Western blot. Gliotoxin cross-links with the target protein via the disulfide bond sulfhydryl group and then inactivates the activity of the heat shock protein Sti1 [8]. Gliotoxin can also serve as a dual inhibitor of farnesyltransferase (IC_50_ 80 µM) and geranylgeranyltransferase I (IC_50_ 17 µM), with pronounced antitumor activity and favorable toxicity profile against breast cancer in vitro and in vivo [91]. Additionally, gliotoxin showed very weak systemic toxicity towards animals in experiments of over 25 mg/kg by subcutaneous injection weekly for 4 weeks compared with control, suggesting the potential of gliotoxin as an antioxidant or antitumor drug-leading compound with a low dose. Moreover, gliotoxin can significantly promote the expression of transforming growth factor (TGF)-β3, leading to the symptoms of shivering, redness around the anus, and bristling of hair in mice [92].

Researchers are engaged in the exploration of novel targets of gliotoxins. Tang et al. reported that gliotoxin can bind to pyruvate kinase, which is a key enzyme for glycolysis and tumor progress; thus, the inhibition of pyruvate via gliotoxin can induce apoptosis in the human gloma cell line U87, synergize with temozolomide. This study provides a novel strategy for gliotoxin-induced apoptosis, thus laying a foundation for the development of gliotoxin as an antitumor drug [93]. It was also reported that gliotoxin can induce cofilin phosphorylation to promote actin cytoskeleton rearrangement through the Cdc42/RhoA-LIMK1 signaling pathway (upregulation of cAMP) in type II human pneumocytes [94,95]; meanwhile, the addition of gliotoxin can promote the invasion of the *gliP* gene deletion of *A. fumigatus* into lung tissue in immunosuppressed mice [96]. Moreover, gliotoxin with a concentration of 20 nM directly disrupted 7SK snRNP by targeting La-related protein 7 (LARP7), releasing active P-TEFb, which phosphorylated the RNA polymerase II C-terminal domain (CTD), thus inducing HIV-1 transcription (upregulating the levels of cell-associated HIV-1 pol RNA copies and reversing HIV-1 latency and ex vivo infected primary CD4^+^ T-cells) without any cytotoxicity [97], thus rendering new clues for the treatment strategy of “activation and shock“, followed by subsequent clearance for HIV disease. Meanwhile, the treatment of 20 nM gliotoxin did not interfere with the activation of CD4^+^ or CD8^+^ T-cells [97], which is important for the elimination of HIV-infection cells. Thus, gliotoxin is also applicable in the diagnosis and treatment of HIV in the future.

## 3. The Prevention of the Toxicity of Gliotoxin and the Producing Strain *A. fumigutas*

Now that the toxic mechanism has been illustrated, the strategies for the prevention of gliotoxin toxicity and the toxicity caused by gliotoxin-producing strain *A. fumigutas* are also summarized in this review. As gliotoxin was synthesized by the *gli* gene cluster in *A. fumigutas* and other fungi, the deletion or deregulation of the essential gene for gliotoxin biosynthesis can be effective for the abrogation of gliotoxin production. Meanwhile, the disulfide bond is critical for the toxicity of gliotoxin; thus, the cleavage of the disulfide bond via different tailoring enzymes is also an ideal alternation for reducing the toxicity of gliotoxin. In addition, the deregulation of positive transcriptional regulators, including pathway-specific regulators and global regulators, can also significantly reduce the production of gliotoxin, thus ameliorating the toxicity of gliotoxin.

### 3.1. The Prevention of the Toxicity of Gliotoxin and A. fumigutas via Biosynthetic Approaches

It has been confirmed that *gli* gene cluster, including *gliP, gliG* and *gliT* genes, is responsible for the biosynthesis of gliotoxin in *A. fumigutas* [16]. The *gliP* gene is responsible for the formation of the dipeptide scaffold; the deletion of *gliP* can significantly attenuate the symptoms of invasive aspergillos caused by *A. fumigutas* via the abrogation of gliotoxin production [42]. Δ*gliP A. fumigatus* showed significantly less virulence towards 129/Sv and BALB/c, which were immunosuppressed by hydrocortisone alone, and this gene deletion strain failed to induce apoptosis towards EL4 thymoma cells via mitochondrial-membrane potential disruption, superoxide production, caspase 3 activation, and phosphatidylserine translocation [98]. Meanwhile, Δ*gliP A. fumigatus* also showed reduced cytotoxicity towards nonneutropenic mice and the drosophila melanogaster model [99]. However, the *glip* deletion *A. fumigatus* strain that cannot produce gliotoxin is still pathogenic towards neutropenic mice that were immune-suppressed by cyclophosphomide combined with hydrocortisone [26,96]. This result suggests the complete immune suppression by hydrocortisone and is an important indicator for the virulence of *A. fumigatus* and that neutrophil is the dominant target for gliotoxin. The supernatant of the *gliP-*deficient strain showed weaker cytotoxicity towards macrophage-like and T-cell lines; however, the *gliP* deletion strain showed no difference in virulence towards completely immunosuppressed mice in a low-dose model compared with wild strain [26], thus further supporting the conclusion that gliotoxin is not required for the pathogenicity of *A. fumigatus* towards completely immunocompromised mice.

It was reported that *gliG-*encoding glutathione-S-transferase (GST) mediates the carbon–sulfur bond formation in gliotoxin biosynthesis and oxygenase GliC mediates the bihydroxylation of diketopiperazine, which is a prerequisite for the glutathione adduct formation catalyzed by GliG [100]. The deletion of *gliG* can completely abolish the production of gliotoxin in *A. fumigutas* and the accumulation of 6-benzyl-6-hydroxy-1-methoxy-3-methylenepiperazine-2,5-dione, thus reducing its virulence [101,102]. The results also revealed that the deletion of *gliC* is also an effective strategy for the abrogation of gliotoxin production in *A. fumigutas*, which needs further confirmation in future studies.

*gliT-*encoding thioredoxin reductase is responsible for the closure of the disulfide bond in gliotoxin. Additionally, the deletion of *gliT* leads to the abrogation of the secretion of gliotoxin and renders the *A. fumigutas* strains highly sensitivity to exogenous gliotoxin [103], which can be alleviated by the addition of glutathione; these results suggest the important role of GliT in the self-protection of *A. fumigutas* towards gliotoxin. Interestingly, *gliT* can also be expressed in the *gliZ* deletion *A. fumigutas* strain by the addition of exogenous gliotoxin, suggesting that *gliT* is independently regulated compared to other genes in the *gli* cluster [103,104]. GliT showed reductase activity towards 9 µM gliotoxin to prevent the irreversible depletion of intracellular glutathione by the oxidized form of gliotoxin, which confers a resistance of gliotoxin to *A. fumigatus* [103]. These investigations suggest the bi-function of GliT as the formation of the disulfide bond in gliotoxin and the reduction of gliotoxin in the presence of high-concentration gliotoxin. Moreover, the transformation of *gliT* can render resistance to exogenous gliotoxin to *Aspergillus nidulans* and *Saccharomyces cerevisiae*, respectively, when transformed with *gliT* [103]. Furthermore, gliotoxin oxidation is impeded in the Δ*gliT* strain, leading to significant S-adenosylmethionine (SAM) depletion and S-adenosylhomocysteine (SAH) overproduction, which can deprive the enzymatic activity of GtmA. This, in turn, results significantly in the accumulation of dithiol gliotoxin, contributing to the hypersensitivity of Δ*gliT A. fumigatus* to exogenous gliotoxin [103]. In addition, the exposure of *A. fumigutas* to exogenous gliotoxin can elevate the expression of genes related to the biosynthesis of gliotoxin, thus facilitating gliotoxin accumulation. Additionally, the proteomic analysis results also indicated a disrupted translation in Δ*gliT A. fumigutas,* thus, in turn, aggravating the sensitivity of Δ*gliT A. fumigutas* to exogenous gliotoxin [103,105].

### 3.2. The Regulation of Gliotoxin by Tailoring Genes

*gtmA-*encoding S-adenosylmethionine (SAM)-dependent bis-thiomethyltransferase, which is not in the *gli* gene cluster, was firstly identified in *A. fumigutas* by in vitro enzymatic assay and gene deletion. In vitro enzymatic assay results indicated that GtmA can convert dithiol gliotoxin to bisdethilobis(methylthio)gliotoxin (BmGT), with much lower cytotoxicity compared with gliotoxin [106], and this conversion can only be observed in *gliT* deletion *A. fumigutas*, suggesting the competitive relationship of GtmA and GliT towards dithiol gliotoxin [107]. The knockout of *gtmA* can completely abrogate the production of BmGT, revealing that GtmA is the only bis-thimethyltransferase in *A. fumigutas.* Therefore, the overexpression of *gtmA* in *A. fumigutas* can attenuate the toxicity of gliotoxin by reducing the production of gliotoxin. Meanwhile, the heterologous expression of *gtmA* can confer the ability of gliotoxin resistance to other hosts. Additionally, the crystal structures of GtmA binding to S-adenosylhomocysteine (1.33 A°) and GtmA complexed to S-adenosylmethionine (2.28 A°) have also been elucidated, thus providing molecular insights to the catalytic mechanism of GtmA towards gliotoxin. The comparison of GtmA-apo and GtmA-SAM crystal structures suggest the important roles of amino acids, including Phe127, Trp157, Asn159, and Trp162, for the catalyzation of GtmA towards gliotoxin and dithiol gliotoxin. Interestingly, GtmA showed much higher catalytic efficiency towards dithiol gliotoxin compared with that of gliotoxin, which was probably due to the cleaved disulfide bridge in dithiol gliotoxin, which is more feasible for the binding of SAM [100]. Furthermore, the crystal structure of the complex of GtmA and S-(5′-adenosyl)-L-homocysteine combined with the molecular docking of gliotoxin and GtmA indicated that the methylation of the C10a-SH group precedes the alkylation of the C3-SH site in dithiol gliotoxin [108].

GliK was proposed to partly remove GSH moities in gliotoxin biosynthesis [16,109]. Gallagher et al. reported that the deletion of the *gliK* gene abrogates the production of gliotoxin and that the Δ*gliK A. fumigutas* strains were much more sensitive to 1 mM H_2_O_2_ and exogenous gliotoxin (10 and 20 μg/mL), suggesting that GliK can serve as a protein that protects *A. fumigutas* from the damage caused by ROS and gliotoxin [110]. Interestingly, the deletion of *gliK* also led to the elevation of ergothioneine [110], which is also an intracellular antioxidant. The results suggest that GliK protein can alleviate the toxicity of gliotoxin and ROS, which led us to hypothesize that the heterologous expression of *gliK* can render resistance to gliotoxin and *A. fumigutas* to different hosts. Moreover, Cys-Gly carboxypeptidase GliJ can act as a dinuclear metallohydrolase in the presence of different metal ions, especially Zn^2+^, to completely remove GSH moieties [109]; thus, the deletion of GliJ can also be taken into consideration to abolish the production of gliotoxin.

GliN was unveiled as an N-methyltransferase that can mediate amide methylation and render stability to gliotoxin and ditholgliotoxin, which act as a substrate for S-methyltransferase [111]. Thus, GliN-mediated N-methyltransferase is the prerequisite for the cytotoxicity of gliotoxin, whereas the GtmA-mediated S-methylation plays an important role in the detoxification of gliotoxin [112]. Therefore, the depletion of *gliN* or the overexpression of *gtmA* can alleviate the production of gliotoxin and promote the detoxification of gliotoxin, respectively.

### 3.3. The Utilization of Toxin Transporter

GliA is the major facilitator of superfamily transporters for the secretion of gliotoxin in *A. fumigutas*; the depletion of GliA would induce the cell death of *A. fumigutas* and significantly attenuate the tolerance of *A. fumigutas* to gliotoxin [113]. The deletion of the *sirA-*encoding ABC transporter resulted in the increased secretion of sirodesmin into the medium, indicating the SirA is not necessary for the production of sirodesmin. *gliA* from *A. fumigatus* can compensate the tolerance of *sirA* deletion *L. maculans* to gliotoxin but not to sirodesmin [114]. The results hint to us that the heterologous expression of the *gliA* gene can improve gliotoxin resistance ability in different hosts, thus ameliorating the hazard of gliotoxins.

### 3.4. The Deletion of Regulator

GliZ has been demonstrated as a positive regulator; the deletion of GliZ in *A. fumigutus* can abolish the expression of other gliotoxin-biosynthesis-related genes and deprive their ability to generate gliotoxin, thus significantly reducing the pathogenicity of *A. fumigutas* [17]. GliZ was reported to bind to the promoters of *gliG*, *gliM*, and *gliN genes* to regulate the biosynthesis of gliotoxin in deep-sea fungus *D. cejpii* FS110 [115]. Furthermore, a novel C_2_H_2_ transcriptional regulator GipA was identified to induce gliotoxin production in the presence of extra copies, and the deletion of gipA resulted in a significantly decreased yield of gliotoxin in *A. fumigutas.* The GipA regulator can bind to the upstream of the *gliA* transporter, which is close to the DNA-binding site of GliZ. Additionally, GliZ is indispensable for the regulation of GliA towards the expression of GliA and GliP. Meanwhile, GlipA is indispensable for the biosynthesis of GliA but not GliP [116]. These findings suggest that GipA can dysregulate the secretion of gliotoxin in *A. fumigutas* via the downregulation of GliA transporter expression. Moreover, another C_2_H_2_ transcription factor, MtfA, involved in the growth and development of *A. fumigutas,* was reported to be related to the expression of *gliZ*; the overexpression of MtfA can enhance gliotoxin production via the upregulation of *gliZ* [117]. In addition, the modification of chromatin can also regulate the production of secondary metabolites. CclA-encoding histone methyltransferase can suppress the production of gliotoxin via the repression of *gliZ* expression. The depletion of *cclA* can significantly enhance the gliotoxin yield via the substantial upregulation of *gliZ* expression [118].

Interestingly, a novel C_2_H_2_ transcription factor, RglT, was identified to regulate the expression of gliotoxin-biosynthesis-related genes, especially *gliT*, via directing binding to the promoter of *gliT*; RglT-depleted *A. fumigutas* is more sensitive to exogenous gliotoxin and oxidative stress as well as the decreased production of gliotoxin and attenuated pathogenicity in a murine model of invasive pulmonary aspergillosis (IPA), suggesting the role of self-protection and the regulation of gliotoxin biosynthesis [119].

VeA served as a regulator of the development in *A. fumigatus*, and the absence of VeA levels led to abnormal conidiation in *A. fumigatus.* Moreover, VeA was also demonstrated to be a positive regulator for gliotoxin, and the deletion of VeA can reduce the production of gliotoxin and protease activity, thus reducing the pathogenicity of *A. fumigatus* [19]. The LaeA regulator was reported to be a global positive regulator for the production of gliotoxin and other toxins. Δ*LaeA A. fumigutas* exhibited much weaker virulence in a murine pulmonary model compared with wild-type *A. fumigutas* and *gliZ-*depleted *A. fumigutas*, indicating that the laeA regulator plays a more important role in the virulence of *A. fumigutas* than the GliZ regulator [18]. FlbA is a regulator of G-protein signaling protein in *Aspergillus* spp. The deletion of the *flbA* gene can promote the cell death and autolysis of *A. fumigutas.* The comparative proteomic analysis of wild-type and Δ*flbA A. fumigutas* revealed that the deletion of the *flbA* gene led to the accumulation of GliT protein and superoxide dismutase (SOD) activity, thereby enhancing the gliotoxin tolerance of *A. fumigutas*, implicating that the FlbA regulator can downregulate GliT expression and SOD activity [120]. Therefore, the FlbA regulator can be over-expressed to attenuate the toxicity of *A. fumigutas* and reduce the production of gliotoxin [121]. MpkA-encoding MAP kinase, which can be activated by iron starvation, was reported to regulate gliotoxin production; the deletion of MpkA can reduce gliotoxin production via the downregulation of *gliN* and *gliT* [122].

### 3.5. Other Microbial or Biosynthetic Strategiesand

dithiol gliotoxin exhibits activity of Zn^2+^ chelation, thus inhibiting Zn^2+^-dependent alkaline phosphatase activity and other Zn^2+^-dependent metalloenzymes via the depletion of intracellular Zn^2+^_._ Proteomic analysis results indicated that excess Zn^2+^ alters the effect of gliotoxin on Δ*gliT A. fumigutas.* The addition of Zn^2+^ can increase the abundance of GtmA due to the disruption of GtmA activity and the chelation of Zn^2+^ by dithiol gliotoxin [123]. Ditholgliotoxin can cause the aggregation of many metalloproteins in vitro, which can be alleviated by the pre-incubation of dithiol gliotoxin with Zn^2+^_._ Thus, the dithiol gliotoxin in vivo can act as a Zn^2+^ chelator to significantly inhibit the growth of *A. fumigutas* in the absence of GliT and GtmA [123]. Additionally, we hypothesize that the addition of excessive Zn^2+^ can facilitate the formation of a Zn^2+^–dithiol gliotoxin complex, thus reducing the production of gliotoxin in *A. fumigutas* and providing a novel strategy for the attenuation of the activity of gliotoxin and *A. fumigutas.*

Interestingly, the depletion of zinc ion can significantly upregulate the expression levels of gliotoxin biosynthesis, especially that of the *gliZ* positive regulator, which is a Zn_2_-Cys_6_ binuclear transcriptional factor; the process may be mediated by a zinc-dependent transcription factor, ZafA, which can be supported by the findings of a presence of ZafA-binding conserved motifs in the upstream region of *gliZ* [124]. The addition of elevated concentrations of zinc ions can inversely decrease the production of gliotoxin, which may be due to the inhibition of *gliZ* via the binding of elevated ZafA, mediated by increased zinc ions [124]. The addition of gliotoxin and zinc chelators can significantly inhibit the growth of Δ*gliT* Δ*gtmA A. fumigutas*, and 1 μM zinc can suppress the biosynthesis of gliotoxin via the dysregulation of the *gliZ* regulator and the chelation of dithiol gliotoxin [111]. These findings enlighten us to employ exogenous dithiogliotoxin to chelate zinc and, thereby, reduce the generation of gliotoxin in *A. fumigutas*; meanwhile, the addition of excessive zinc ions can inversely attenuate the pathogenicity of *A. fumigutas* to some extent via the reduced production of gliotoxin.

It was reported that the existence of lipopolysaccharides, peptidoglycans, or lipoteichoic acid in the growth media, at a concentration of 5 mg/mL, increased the gliotoxin concentration of *A. fumigutas* in the media by 37%, 65%, and 35%, respectively [125], thus offering a microbial strategy to enhance the biosynthetic efficiency of gliotoxin in *A. fumigutas*. Conversely, the outcomes also indicated that the production of gliotoxin in *A. fumigutas* via the reduction of the production of lipopolysaccharides, peptidoglycans, or lipoteichoic acid.

In all accounts, the biosynthetic approaches for the attenuation of the toxicity caused by gliotoxins are summarized in Figure 5, thus providing effective biosynthetic strategies for the alleviation of the production and toxicity of gliotoxin in *A. fumigutas.*

## 4. Conclusions and Prospects

Taken together, gliotoxin can show cytotoxic effects via antiangiogenesis, inflammation, immunosuppression mediated by the NF-κB and DNA damage, peroxide damage mediated by production of ROS mediated by the inhibition of NADPH oxidase and redox cycling, and apoptosis via various signal pathways, including caspase-dependent mitochondrial membrane potential disruption, the NF-κB pathway and NOTCH_2_-dependent apoptosis. In contrast, gliotoxin can also be beneficial in low doses. Low-dose gliotoxin can be employed as an antioxidant to mitigate ROS damage in the presence of the thioredoxin redox system. Moderate doses of gliotoxin also exhibit an anti-inflammatory effect in vivo via the suppression of NF-κB activity. To our surprise, gliotoxin with a dose of less than 40 nM can activate latent HIV-1 gene expression, thus facilitating the diagnosis and treatment of HIV.

Moreover, biosynthetic approaches to alleviate the toxicity of gliotoxin and the pathogenicity of *A. fumigutas* are also summarized in this review. The deletion of indispensable biosynthetic genes for gliotoxin, including *gliP*, *gliG*, *gliC*, *gliJ*, and *gliN*, can abolish the production of gliotoxin and attenuate the pathogenicity of *A. fumigutas* to some extent. *gliT* exerts the bifunction of the formation of a disulfide bond in gliotoxin and a self-protection role by the reduction of excessive gliotoxin in *A. fumigutas*; GliA act as a transporter for the secretion of gliotoxin, thus reducing the toxicity of gliotoxin towards various hosts; the expression of GliA can be regulated by a GipA regulator. GtmA plays a role in the S-methylation of ditholgliotoxin, thus reducing the toxicity of gliotoxin. GiZ is a positive regulator for the biosynthesis of gliotoxin, which can be dysregulated by excessive zinc through the activation of the zinc-dependent regulator ZafA that binds to the GliZ regulator.

In conclusion, gliotoxin can induce cytotoxic effects with relatively high concentrations (more than 100 nM); alternatively, low doses of combined gliotoxins can also be developed as a potential HIV therapeutic agent and as an antioxidant in the presence of a thioredoxin agent. Gliotoxins have also been used as antibiotics in the control of plant pathogens, including *Pythium ultimum* and *Sclerotinia sclerotiorum*. Furthermore, low doses of gliotoxin can also serve as immune inhibitors in organ-transplant patients as well as an antitumor leading compound via the combination of some targeting strategies, such as RGD-peptide-conjugated exosomes and nanoparticles, thus promoting the development of this biomedical field.

## Figures and Tables

**Figure 1 ijms-22-13510-f001:**
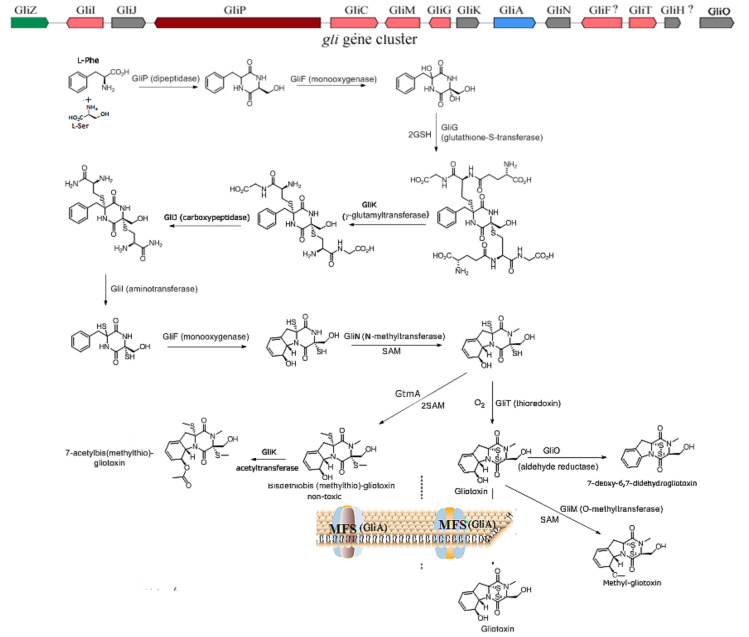
The proposed biosynthetic pathway of gliotoxin and its derivatives [16]. The *gli* gene cluster depicts the biosynhetic pathway of gliotoxin in *A. fumigutas,* the coding protein for each gene was annoated in the biosynthetic pathway of gliotoxin and thereof derivatives. The SAM refers to S-adenosylmethionine, MFS refers to major facilitator superfamily. GtmA refers to S-methyltransferase.

**Figure 2 ijms-22-13510-f002:**
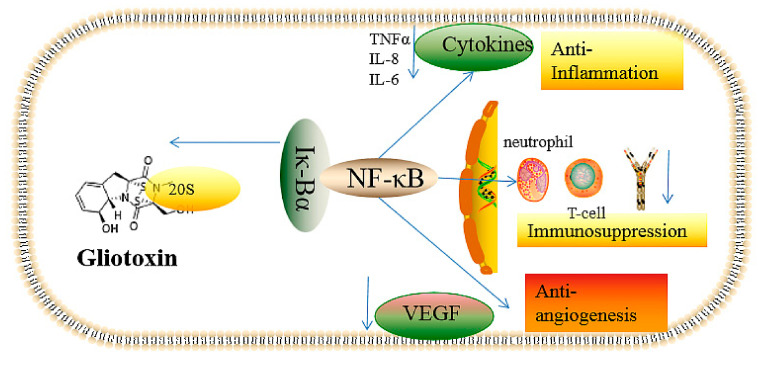
The cytotoxic effects, including immunosuppression, antiangiogenesis, and anti-inflammation, of gliotoxin mediated by NF-κB. The arrow to the left indicates that gliotoxin can detach 20S protease from Iκ-Bα, thus inhibiting the activation of NF-κB. The downward arrow indicates the down-regulation caused by the inactivation of NF-κB mediated by gliotoxin.

**Figure 3 ijms-22-13510-f003:**
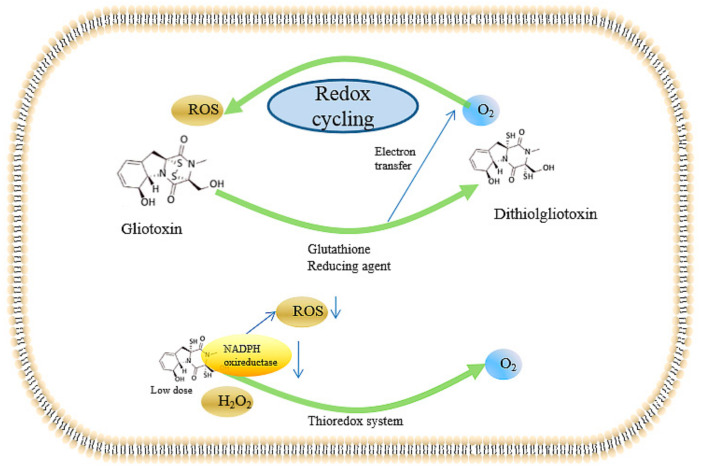
Gliotoxin-mediated redox cycling system and the antioxidant effect. The downward arrow indicates the ROS content was decreased by low dose of gliotoxin via the inhibition of NADPH oxireducatase activity in the presence of thioredox system.

**Figure 4 ijms-22-13510-f004:**
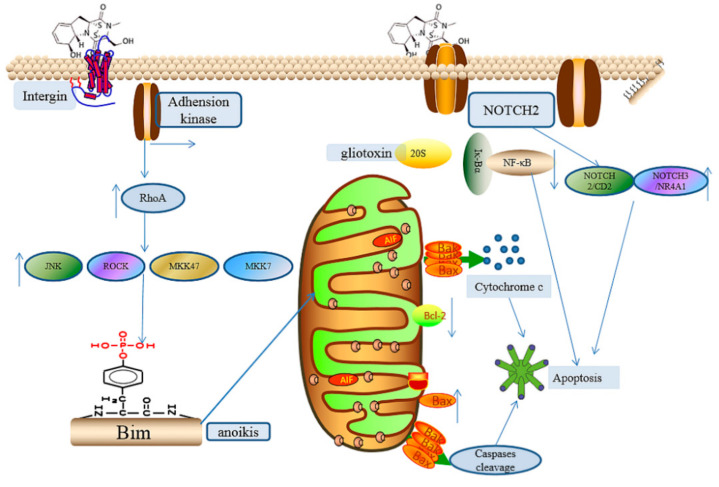
Gliotoxin-mediated apoptosis via different signal pathways. The arrows besides the specific protein indicates the alternation of expression levels of target proteins caused by gliotoxin. For example, the upward arrow of RhoA indicates that the expression level of RhoA is up-regulated by gliotoxin treatment.

**Figure 5 ijms-22-13510-f005:**
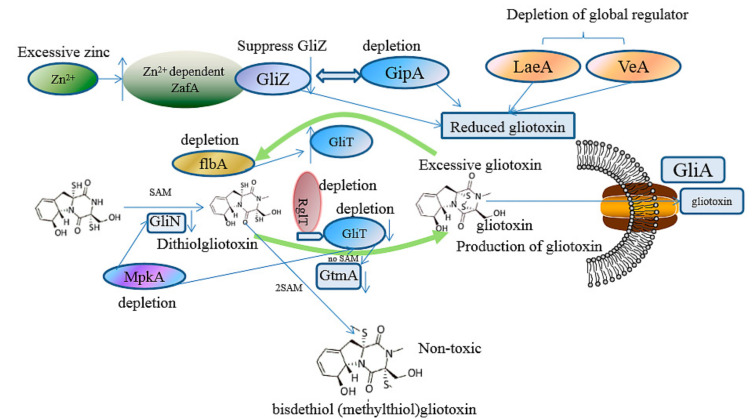
The detoxification mechanism via biosynthetic strategies in *A. fumigutas.* The downward arrow indicates the depletion of specific gene leads to the down-regulation of target gene. For example, the deletion of MpkA regulator results in the down regulation of GliT, and the knockout of *gliT* leads to the depletion of SAM, thus contributing to the enzymatic activity loss of GtmA.

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
