# Peer review of "The Toxic Mechanism of Gliotoxins and Biosynthetic Strategies for Toxicity Prevention"

_ijms, 2021, doi:10.3390/ijms222413510_

Round 1
Reviewer 1 Report
The authors provide a very comprehensive review of the toxic effects of gliotoxins in order to prevent this toxicity.
The review is well prepared (with the exception of ew typings listed after), and will be for sure a very useful review for specialists but also for applied researchers.
I recommend this paper for publication following the correction of the minor typing mistakes and the addition of a figure showing gliotoxin structure as well as its biosynthetic pathway would be useful.
Minor changes:
use italics for genes and font for proteins (eg: gliN line 71 should be in italics)
missing space: for example Line 102: vitro[20, 23], Line 96 antibiotics[21].
Author Response
Responses to Reviewer’s Comments
Reviewer 1#
The authors provide a very comprehensive review of the toxic effects of gliotoxins in order to prevent this toxicity.
The review is well prepared (with the exception of few typings listed after), and will be for sure a very useful review for specialists but also for applied researchers.
I recommend this paper for publication following the correction of the minor typing mistakes and the addition of a figure showing gliotoxin structure as well as its biosynthetic pathway would be useful.
Our response:
Thank you very much for your comments. We have added Figure 1 which proposed the biosynthetic pathway of gliotoxin and thereof derivatives.
Minor changes:
use italics for genes and font for proteins (eg: gliN line 71 should be in italics)
Our response:
We have corrected genes to italics now.
missing space: for example Line 102: vitro[20, 23], Line 96 antibiotics[21].
Our response:
The spaces have been added between literature and words now.
Reviewer 2 Report
Gliotoxin is the most famous of the family of epipolythiodioxopiperazine alkaloids. There are many reports concerning its biological activity and the authors performed a good summary of the literature. They have organized the material according to the type of activity and the cellular pathways involved. Overall, this is a nice review. My only suggestions are the following:
- Is gliotoxin itself biologically active, or a prodrug for the dithiol form? The identity of the active agent should be clarified.
- Is there information on how gliotoxin (or the dithiol) interact with the identified targets e.g. is it reversible or irreversible inhibition, and is there structural data of protein-drug complexes?
- Some discussion of other members of the epipolythiodioxopiperazine alkaloids should be included. How do they compare in terms of potency and ease of production to gliotoxin? Are they similar in their cellular targets or different from gliotoxin?
Author Response
Responses to Reviewer’s Comments
Reviewer 2#
Gliotoxin is the most famous of the family of epipolythiodioxopiperazine alkaloids. There are many reports concerning its biological activity and the authors performed a good summary of the literature. They have organized the material according to the type of activity and the cellular pathways involved. Overall, this is a nice review. My only suggestions are the following:
1. Is gliotoxin itself biologically active, or a prodrug for the dithiol form? The identity of the active agent should be clarified.
Our response:
Thank you for your opinion. Both the gliotoxin and dithiol gliotoxin are cytotoxic, however, gliotoxin is more toxic probably due to the existence of disulfide bridge, which can bind to many physiological related proteins. This was added in the section of 2.2.4.
2. Is there information on how gliotoxin (or the dithiol) interact with the identified targets e.g. is it reversible or irreversible inhibition, and is there structural data of protein-drug complexes?
Our response:
Thank you for your comments. The crystal structures of complex comprising GtmA and SAM, SAH have been reported, the molecular docking of GtmA and gliotoxin has also been proposed, indicating the key amino acids for the biochemical reactions of GtmA towards gliotoxin. These sentences have been added in section 3.2.
3. Some discussion of other members of the epipolythiodioxopiperazine alkaloids should be included. How do they compare in terms of potency and ease of production to gliotoxin? Are they similar in their cellular targets or different from gliotoxin?
Our response:
Thank you for your comments. The biosynthetic pathway of gliotoxin and thereof derivatives is proposed in our revised manuscript. We have isolated gliotoxin and its derivatives from deep sea fungus Dichotomyces cejpii FS110 and Geosmithia pallida FS140 with relatively high yield. And epipolythiodioxopiperazine derivatives of gliotoxin with dithiol group showed weaker cytotoxicity compared with gliotoxin, and gliotoxin derivate with methylthio group showed much weaker cytotoxicity than that of dithiolgliotoxin, furthermore, the depletion of methythio group in gliotoxin derivative lead to the loss of cytotoxicity, suggesting the important role of disulfide bridge and thiol group in the biological activity of gliotoxin and its derivatives. This sentence has been added in the section “2.1".